



# Implementation of aerosol data assimilation in WRFDA (V4.0.3) for WRF-Chem (V3.9.1) using the MADE/VBS scheme

Soyoung Ha

National Center for Atmospheric Research, Boulder, Colorado, USA

**Correspondence:** Soyoung Ha (syha@ucar.edu)

**Abstract.** The Weather Research and Forecasting model data assimilation (WRFDA) system, initially designed for meteorological data assimilation, is extended for aerosol data assimilation for the WRF model coupled with Chemistry (WRF-Chem). An interface between WRF-Chem and WRFDA is built for the Regional Atmospheric Chemistry Mechanism (RACM) chemistry and the Modal Aerosol Dynamics Model for Europe (MADE) coupled with the Volatility Basis Set (VBS) aerosol schemes.

This article describes the implementation of the new interface for assimilating $PM_{2.5}$, $PM_{10}$, and four gas species ($SO_2$, $NO_2$, $O_3$, and co) on the ground. And the effects of aerosol data assimilation are briefly examined through a month-long case study during the Korea-United States Air Quality (KORUS-AQ) period. It is demonstrated that the 3DVAR analysis can lead to consistent forecast improvements up to 26%, diminishing most systematic bias errors for 24 h.

## 1  Introduction

Regional air quality forecasting is mainly concerned with air pollutants confined to the boundary layer (up to 1 km from the ground), which is predominantly characterized by near-surface concentrations of particulate matter (PM) with particle diameters less than 2.5 $\mu$m and 10 $\mu$m (e.g., $PM_{2.5}$ and $PM_{10}$, respectively). Many processes, such as the transport and dispersion of chemical species, that directly affect surface concentrations, strongly depend on weather conditions (Baklanov et al., 2017). In particular, frequent haze events with high PM concentrations over Korea are often associated with long-

range transport of pollutants so that the changes in local emissions in the upstream areas can affect the chemical composition and the PM concentrations in the region to a great extent (Jo et al., 2020). Given the considerable effect of meteorological simulations on the chemical processes and the large uncertainty in the chemical transport models, chemical data assimilation in the numerical weather prediction (NWP) system coupled with chemistry can make encouraging contributions to short-range air quality forecasting, with better representation of the atmospheric composition at the initial time.

The Weather Research and Forecasting model's community data assimilation system (WRFDA; Barker et al. (2012)) developed by the National Center for Atmospheric Research (NCAR) was initially designed for meteorological data assimilation to initialize the WRF model (Skamarock et al., 2008) using a variational or a hybrid data assimilation technique. Since the NWP model coupled with chemistry and aerosol dynamics as an integrated forecasting system (WRF-Chem; Grell et al. (2005)), it has been widely used for regional air quality forecasting, representing two-way real-time interactions between meteorology

and chemistry (Grell and Baklanov, 2011), but WRFDA has remained for weather data assimilation until very recently.





As most prognostic variables (except hydrometeors) remain the same regardless of the physics parameterization employed in the WRF model, it is common practice to compare different physics schemes such as planetary boundary layer (PBL) or microphysics, trying to find an optimal configuration for a particular region or case. On the contrary, each chemical option (for both gas and aerosol chemistry) defines a unique set of chemical and aerosol prognostic variables in the WRF-Chem model. Therefore, aerosol data assimilation using variational techniques requires developing a new interface for the particular chemistry option (*chem_opt*), and the analysis system gets tied to the chemical parameterization. Moreover, anthropogenic emission data also need to be produced for the chemistry variables defined in the same chemical option. That is why chemical or aerosol data assimilation studies have used a minimal set of chemical options and why it is challenging to make a clean comparison between different aerosol schemes in the variational analysis framework.

Other challenges for chemical data assimilation are large model uncertainties due to the complexity of chemical processes, highly nonlinear and non-Gaussian error distribution of chemical species, and expensive computations ascribed to a long list of chemical species - typically with dozens or hundreds of prognostic variables. The latter makes the three-dimensional variational data assimilation (3DVAR) algorithm still attractive, especially in the operational environment, even with its strong limitations such as static background error covariance and the use of the linearized forecast model during the minimization procedure.

Due to the simplicity and the effective cost, the Goddard Chemistry Aerosol Radiation and Transport (GOCART; Chin et al. (2002)) aerosol scheme has been commonly used for aerosol data assimilation (Liu et al. (2011), Saide et al. (2014), and Ha et al. (2020), just to name a few), but it is well known to underestimate ground PM concentrations due to the lack of secondary organic aerosols (SOA) formulation (McKeen et al. (2009), Pang et al. (2020)).

To use a more sophisticated chemistry option in aerosol data assimilation, Sun et al. (2020) implemented a new interface between WRF-Chem and WRFDA for the Carbon Bond Mechanism version Z (CBMZ; Zaveri1999) gas chemistry and Model for Simulating Aerosol Interactions and Chemistry (MOSAIC; Zaveri et al. (2008)) aerosol schemes. They assimilated surface measurements using the 3DVAR technique and demonstrated systematic improvements of air quality forecasting over China up to 24 h.

This study extends the WRF-Chem/WRFDA 3DVAR system for the Regional Atmospheric Chemistry Mechanism (RACM; Stockwell et al. (1997)) gas-phase chemistry, coupled with the Modal Aerosol Dynamics Model for Europe (MADE; Ackermann et al. (1998)) and the secondary organic aerosol (SOA) scheme based on a four-bin volatility basis set (VBS) (Ahmadov et al., 2012) in version 3.9.1 of the WRF-Chem model. This chemical option is expected to provide a more realistic representation of organic carbon (OC) and SOA that often resulted in the $PM_{2.5}$ underestimation over the East Asian region (Lee et al., 2020; Jo et al., 2020).

The goal of the new system development is to facilitate aerosol data assimilation using the RACM-MADE-VBS scheme (chem_opt = 108) so that the aerosol initial conditions can lead to better air quality forecasting, especially in surface $PM_{2.5}$ concentrations over South Korea. This study demonstrates the assimilation capability of the WRF-Chem/WRFDA using the parameterization and examines how the assimilation of surface observations can affect air quality forecasts through month-long cycling experiments for May 2016, the Korea-United States Air Quality (KORUS-AQ) campaign period. For the details of the





case or the field campaign, readers refer to several other papers (Ha et al. (2020), Peterson et al. (2019), and Miyazaki et al. (2019)).

An overview of the new WRF-Chem/WRFDA system, including new forward operators and background error statistics, is presented in Section 2, followed by cycling experiments and the forecast verification against independent observations described in Section 3. Finally, conclusions are made in Section 4, followed by a discussion on the limitations of this study and

suggestions for future research.

## 2   The WRF-Chem analysis and forecasting system

The WRF-Chem model has many options for gas and aerosol chemistry parameterizations that are fully coupled with meteorology, facilitating aerosol direct and indirect effects through interactions with radiation, photolysis, and microphysical processes in real-time (Fast et al., 2006). Within the WRF infrastructure, it is coupled with the WRF preprocessing system (WPS) and

data assimilation (WRFDA), so it can fully support the analysis and forecast cycling with the coupled evolution of weather and chemistry. As the processes like advection and diffusion are applied for both chemical and meteorological variables, chemical species are transported based on the model time step (which depends on the grid resolution).

The variational data assimilation pulls the model trajectories toward the observed information at the initial time, but once the model integration starts from the initial state, the forecast model tries to restore its own climatology based on the assumptions

and parameters defined in the system. Hence, in order to prevent the model state from drifting away from the observed state, data assimilation (or the analysis) incorporates various observations into the model, updating initial conditions at certain time intervals (ex. every 6 h). By conducting the analysis and the forecast consecutively (so-called $cycling$), initial conditions and subsequent forecasts can be systematically improved in the long term.

By default, chemical boundaries are reset based on the idealized profiles specified in the chemistry routines in WRF-Chem.

But if the chemical lateral boundary option (e.g., "$have\_bcs\_chem$") is turned on to provide more realistic inflows of chemicals, wrfbdy files also need to be updated for chemical species, typically using the output from global chemical forecasts.

Unlike the cycling for weather forecasting with WRF, chemical simulations typically recycle chemistry fields in the forecasts from the previous cycle, which are provided through an auxiliary input stream for the initialization (with $real.exe$). Also, the WRF-Chem needs several additional input files for various emissions data because the chemical transport model is strongly

driven by the forcing parameters throughout the model integration and heavily relies on the quality of the emissions data produced for the region.

Once the initial conditions (e.g., wrfinput files) are produced, they are used as background (e.g., $a\ priori$) for the analysis at the cycle. In this study, chemical observations are assimilated as well as $in\ situ$ meteorological measurements. But even if weather data are not assimilated, meteorological fields are updated through initial and boundary conditions and the online

interactions between aerosol and radiation during the forecast step. Once the assimilation is done, wrfbdy files are updated in the mother domain before the model forecast starts, to become consistent with the analysis fields in the relaxation zone. But



such boundary updates are not applied to chemical fields because chemical or aerosol observations within five grids ($5\Delta x$) from the boundary cells are not assimilated (e.g., no analysis updates near the lateral boundaries).

## 2.1 The WRF-Chem configuration

The model simulations cover the East Asian region and the Korean peninsula with 27- and 9-km grid resolution, respectively, in a one-way nesting mode, as shown in Fig. 1. Vertically, 31 model levels are configured up to 50 hPa, with the lowest level located around 173 m in domain 2. Such a coarse vertical resolution may not resolve the observed spatial and temporal variability of atmospheric aerosols, but the configuration is adopted from the current operational setting in the National Institute of Environment Research (NIER) in South Korea.

The static geographical fields such as land use, vegetation fraction, albedo, soil temperature, and moisture are obtained from the 20-class, 30 arc-second MODIS data through the geogrid program of the WRF preprocessing system (WPS). The initial and lateral boundary conditions for meteorological variables are produced by global forecasts from the UK Met Office's Unified Model (UM) operated by the Korean Meteorological Administration (KMA) every 6 h. For meteorological data assimilation, conventional observations in the National Centers for Environmental Prediction (NCEP) prepbufr data

(https://rda.ucar.edu/datasets/ds337.0/; last access: 4 Mar 2021) are employed.

This study focuses on the RACM gas chemistry and the MADE-VBS aerosol parameterization (e.g., chem_opt = 108) in WRF Version 3.9.1. The MADE-VBS aerosol scheme defines a superposition of three log-normal modes - Aitken, accumulation and coarse modes - based on the particle size distribution: an Aitken mode with a median diameter of 0.01 $\mu$m, an accumulation mode ranging between 0.01 and 1 $\mu$m, and a coarse mode for particles typically larger than 1 $\mu$m (with a median

around 10 $\mu$m). All aerosol particles are assumed to be spherical and internally mixed (Aquila et al., 2011). The aerosol species treated are sulfate ($SO_4^=$), nitrate ($NO_3^+$), ammonium ($NH_4^+$), elemental carbon (EC), primary organic matter (POA), anthropogenic and biogenic secondary organic aerosol (SOA), chloride (Cl), sodium (Na), unspeciated $PM_{2.5}$, unspeciated coarse fraction of $PM_{10}$ (antha), soil dust, and sea salt. The unspeciated $PM_{2.5}$ includes the fine fraction of sea salt and mineral dust aerosols.

The dust and sea salt emissions are simulated following the GOCART mechanism (e.g, dust_opt = 13 and seas_opt = 2). Photolysis rates of chemical species are computed in a simplified version of the National Center for Atmospheric Research (NCAR) Tropospheric Ultraviolet-Visible (TUV) model (named the Madronich scheme) (Madronich, 1987). And the Rapid Radiative Transfer Model (RRTMG) is used for both short-wave (ra_sw_physics=4) and long-wave (ra_lw_physics=4) radiation. The direct aerosol effect is accounted for through interactions with atmospheric radiation and photolysis. A list of physics

and chemistry schemes used in this study is summarized in Table 1.

Anthropogenic emission data are obtained from the KORUS version 2 inventory, originally developed based on the Comprehensive Regional Emissions for Atmospheric Transport Experiment (CREATE-2015) emissions dataset and updated for the KORUS-AQ campaign (Woo et al., 2012; Choi et al., 2019). They were all emitted at the surface, i.e., without any plume rise or specified vertical distribution. Biogenic emissions are built up online using the Model of Emission of Gases and Aerosol

from Nature (MEGAN; Version 2) (Guenther et al., 2006), but biomass burning emissions are not used in this study. All the





WRF files including anthropogenic and biogenic emissions are processed based on the MODIS landuse datasets (Friedl et al., 2002).

For chemical lateral boundary conditions, 6-hourly global outputs from the Community Atmosphere Model with Chemistry (CAM-Chem) model, a component of the Community Earth System Model (CESM) version 2.1, were used (Buchholz et al.,
2019). These simulations were configured at 0.9x1.25° horizontal resolution and 56 vertical levels up to 1.9 hPa using an updated tropospheric chemistry mechanism (MOZART-T1; Emmons et al. (2020)), the Modal Aerosol Model with 4 modes (MAM4; Liu et al. (2016)), the anthropogenic and biomass burning emissions from the inventories specified for Climate Model Intercomparison Project 6 (CMIP6), and meteorological fields specified from Modern-Era Retrospective analysis for Research and Applications (MERRA)-2 reanalysis (Molod et al., 2015). To make chemical boundary conditions for domain 1, chemical
species in CAM-Chem are converted to the RACM gas species in WRF-Chem through the "mozbc" utility (downloaded from https://www2.acom.ucar.edu/wrf-chem/wrf-chem-tools-community/, last access: 28 December 2020).

## 2.2 WRFDA for WRF-Chem

A new interface for the RACM-MADE-VBS scheme is developed based on version 4.0.3 of the WRFDA system to assimilate surface $PM_{2.5}$, $PM_{10}$, $so_2$, $no_2$, $o_3$, and $co$ measurements. The variational data assimilation system seeks an analysis solution
as the best estimate of the true state by minimizing deviations of model variables ($\mathbf{x}$) from the corresponding observations ($\mathbf{y}$) based on the error statistics of background forecasts and observations. The variational scheme assumes Gaussian and unbiased error distributions, which can be characterized by covariances alone, its solution is thus found a least-squares best fit using the covariances. In practice, when the cost function $J(\mathbf{x})$ is reached to a minimum through an iterative minimization process, the resulting state vector $\mathbf{x}$ becomes the analysis solution (Lorenc, 1986).

$$J(\mathbf{x}) = J^b + J^o = \frac{1}{2}(\mathbf{x} - \mathbf{x_b})^{\mathbf{T}}\mathbf{B^{-1}}(\mathbf{x} - \mathbf{x_b}) + \frac{1}{2}(\mathbf{y} - H(\mathbf{x}))^{\mathbf{T}}\mathbf{R^{-1}}(\mathbf{y} - H(\mathbf{x})), \tag{1}$$

where $\mathbf{x_b}$ is a deterministic forecast from the previous assimilation cycle with subscript $\mathbf{b}$ denoting background forecasts, and $\mathbf{B}$ and $\mathbf{R}$ represent background and observation error covariance matrices, respectively. An observation operator $H(\mathbf{x})$ transforms model states ($\mathbf{x}$) to observed quantities ($\mathbf{y}$) at observation locations and can be nonlinear.

The WRFDA employs an incremental formulation (Courtier et al., 1994) where observations are assimilated through the
observation operator to provide analysis increments $\delta\mathbf{x}(= \mathbf{x} - \mathbf{x_b})$ that minimizes $J$. This approach can keep analysis imbalance to a minimum, making the minimization procedure more efficient. The resulting analysis $\mathbf{x_a}$ ($= \mathbf{x_b} + \delta\mathbf{x}$) is then used as the initial condition for the following forecast. For a typical numerical weather prediction (NWP) model, the state vector ($\mathbf{x}$) that contains all the prognostic variables lies in the huge dimensional state space (with typical degrees of freedom greater than $O(10^7)$), which makes the computation of $J^b$ prohibitive. As a practical way of solving $J^b$, a control vector ($\mathbf{v}$) that consists of
analysis variables is defined as $\delta\mathbf{x} = \mathbf{B^{1/2}}\mathbf{v}$. While forecast errors of model variables are typically correlated through governing equations, control variables are designed to have no cross-correlations such that the error matrix is diagonalized. With the





control variable transformation, the cost function is rewritten as below.

$$J(\mathbf{v}) = \frac{1}{2}\mathbf{v^T}\mathbf{v} + \frac{1}{2}(\mathbf{d} - \mathbf{HB^{1/2}v})^\mathbf{T}\mathbf{R^{-1}}(\mathbf{d} - \mathbf{HB^{1/2}v}),\tag{2}$$

where the innovation vector is defined as $\mathbf{d} = \mathbf{y} - \mathbf{H(x_b)}$ and $\mathbf{H}$ is a linearized version of $H$. In weather data assimilation, the

control variable transformation has been broadly practiced because meteorological variables follow physical balance equations (such as geostrophic or hydrostatic equations) at large scales (Bannister, 2008). But it is not straightforward to define multivariate correlations between chemical species or between chemical and meteorological variables due to their complex interactions and chemical reactions that are highly nonlinear and often transient. Therefore, chemical or aerosol species in the model states ($\mathbf{x}$) are directly used as control variables ($\mathbf{v}$) in chemical data assimilation. For the MADE-VBS aerosol scheme, the analysis

(or control) variables consist of 35 aerosol species in the three-dimensional model space.

To compute the background error covariance matrix ($\mathbf{B}$) for atmospheric constituents, the GEN_BE v2.0 (Descombes et al., 2015) software is expanded for 39 three-dimensional chemical variables: aerosol sulfate (so4ai and so4aj), nitrate (no3ai and no3aj), ammonium (nh4ai and nh4aj), chloride (clai and claj), primary organic matter (orgpai and orgpaj), elemental carbon (eci and ecj), sodium (naai and naaj), unspeciated PM$_{2.5}$ (p25ai and p25aj), 4-bin anthropogenic and biogenic SOA (asoa1i,

asoa1j, asoa2i, asoa2j, ..., bsoa4i, bsoa4j) with $i$ and $j$ at the end of each variable name indicating Aitken and accumulation mode, respectively. Also included are three coarse-mode variables - non-reactive anthropogenic aerosol (antha), marine aerosol concentration (seas), soil-derived aerosol particles such as dust (soila) - and four gas species (so$_2$, no$_2$, o$_3$, and co).

In the variational algorithms, the square root of the $\mathbf{B}$ matrix ($\mathbf{B} = \mathbf{B^{1/2}}(\mathbf{B^{1/2}})^\mathbf{T}$) is decomposed into a series of sub-matrices for the control variable transform.

$$\mathbf{B^{1/2}} = \mathbf{U_p}\mathbf{S}\mathbf{U_v}\mathbf{U_h}\tag{3}$$

where the $\mathbf{U_p}$ matrix is called physical or balance transformation (via linear regression), $\mathbf{S}$ a diagonal matrix of forecast error standard deviation, $\mathbf{U_v}$ the vertical transform, and $\mathbf{U_h}$ the horizontal transform matrix.

The WRFDA provides various options for estimating the background error covariance through "cv_option" in namelist. Here, cv_option = 7 is chosen for no balance transformation in the regional simulations, meaning that the chemical species

are control variables as full fields such that $\mathbf{U_p}$ becomes an identity operator for chemical data assimilation. The horizontal transform matrix $\mathbf{U_h}$ is performed using recursive filters (Purser et al., 2003), while the vertical transform $\mathbf{U_v}$ is carried out via an empirical orthogonal function (EOF) decomposition of the vertical component of the background error covariance.

In the 3DVAR algorithm, background error covariance estimates are important, particularly in data-sparse regions. As most surface stations are concentrated in urban areas, the background error covariance can play a major role on spreading out the

observed information horizontally and vertically.

In this study, chemical simulations are carried out in the WRF-Chem model, starting at 00 UTC every day for one month of May 2016, to compute background error covariance statistics for chemical and aerosol species defined in the RACM-MADE-VBS parameterization. Differences between 24- and 48-h forecasts at the same validation time are then used as a proxy for forecast errors in each domain, and a total of 29 sample forecasts for 3 - 31 May 2016 were used to construct the $\mathbf{B}$ matrix using





the National Meteorological Center (NMC) method (Parrish and Derber, 1992), assuming the same model bias and uncorrelated model errors. There are 5 sequential stages (e.g., stage0 - stage4) implemented with different options in the GENBE software. In this study, all the grid points are binned together for each model level, with no latitudinal or longitudinal dependencies in the background error covariance.

     Figure 2 displays the vertical profile of the background error standard deviation (S) of each species over domain 2. During

the analysis procedure, the error standard deviation is used to weigh the analysis increment for a given variable, affecting how much the observed information will change the model variable. Depending on the aerosol size distribution, Aitken, accumulation, and coarse mode variables are compared separately. Most aerosol species in the accumulation mode have relatively large background error standard deviations in the boundary layer, and their counterparts in the Aitken mode show one order magnitude smaller values, mostly with the maximum at the surface. Among the species, large standard deviations are found

in sulfate, nitrate, ammonuim, and unspecified $PM_{2.5}$, contributing most to $PM_{2.5}$ concentrations. In the coarse mode, sea salt linearly reduces with height, indicating large uncertainties at the sea level, but soila is characterized by the high peak in the mid-troposphere, which might be associated with the large uncertainty in the long-range transport of dust aerosols. In the gas species, ozone represents most uncertainties near the top (e.g., low stratosphere), while carbon monoxide shows large error values in the low troposphere. The vertical error structure is hard to see in $so_2$ and $no_2$ due to the trivial values, but their

standard deviations are also relatively large in the boundary layer.

     The vertical spread of the observed information at the surface is determined by vertical error correlations, closely associated with the simulated boundary layer height. As the static background error covariance cannot simulate the diurnal variability of the boundary layer, this becomes one of the main limitations of the 3DVAR analysis for air quality applications. Figure 3 depicts the normalized vertical auto-correlations derived from the time-lagged forecasts for four major aerosol species in

accumulation and Aitken modes, three coarse mode aerosols, and four trace gases (from top to bottom panels). Generally, correlation contours tend to spread more in the lower levels, implying that the updates in the lowest level can affect the entire boundary layer. The accumulation and coarse mode particles have a wider vertical spread than the Aitken mode particles that would have more localized effects. The circular pattern around level 22 in most species could be related to the advection with strong upper-level jets. While all the trace gases have relatively large correlations near the surface, ozone and nitrogen dioxide

show the largest correlations near the tropopause and stratosphere, respectively.

     To examine the horizontal propagation of the increments from point observations, the horizontal correlation length scales of the same species are illustrated in Fig. 4. In accumulation and coarse modes (in the top and the third rows, respectively), the overall vertical structure is similar, with the linear increase down to the surface. The length scale at the surface is specified around 36 km for so4aj, for example, corresponding to four grids in the 9-km domain, meaning that an observation at a point

location can affect four surrounding grid points radially. On the other hand, Aitken mode variables have short length scales near the surface, which tend to increase in the upper levels, but their maximum values are smaller than those of their counterparts in the accumulation mode, representing more localized effects horizontally. Trace gases show different vertical distributions with the maximum near the top, except for ozone.





When the RACM-MADE-VBS option (e.g., chem_opt = 108) is chosen, the model equivalent of the observed PM$_{2.5}$ is
computed as a total sum of three-dimensional mass mixing ratios of 32 aerosol species in accumulation (j) and Aitken (i)
modes predicted in the WRF-Chem model, as below.

$$\pi_{pm_{2.5}} = \rho_d \sum_{p=1}^{N} \sum_{m=i}^{j} X_m^p, \tag{4}$$

where $\rho_d$ is dry density ([kg/m$^3$]) for the unit conversion from aerosol mixing ratios ($\mu$g/kg) to mass concentrations ($\mu$g/m$^3$),
and N = 16. The observation operator follows the way the MADE-VBS scheme estimates PM$_{2.5}$ concentrations in the model,
having individual species in different modes contributing to PM concentrations equally. When the PM observations are as-
similated, innovations ((o-f)'s) are computed in PM concentrations. Then, once the minimization is completed, the individual
species are updated proportionally to the PM component fraction (e.g., 1/N). If the observed atmospheric composition signifi-
cantly differs from the one in the model, or particular species change predominantly, this approach can lead to erroneous results
(in both analysis and forecast).

When PM$_{10}$ is assimilated alone, the model correspondent is computed by adding three coarse-mode variables - anthro-
pogenic primary aerosol (antha), marine aerosol concentration (seas), soil-derived aerosol particles such as dust (soila) - into
the simulated PM$_{2.5}$. But in the concurrent assimilation of PM$_{10}$ and PM$_{2.5}$, the residuals from (PM$_{10}$ - PM$_{2.5}$) are assimilated
as a sum of three coarse-mode aerosols, following Pang et al. (2018) and Sun et al. (2020).

### 2.3  Observation processing and measurement errors

In this study, hourly surface observations of six major pollutants (PM$_{2.5}$, PM$_{10}$, so$_2$, no$_2$, o$_3$, and co) are used from 379
Korean sites operated by the NIER (http://www.airkorea.or.kr, last access: 21 December 2020) and around 1600 sites from
the China National Environmental Monitoring Center (CNEMC; http://www.cnemc.cn, last access: 21 December 2020) during
the KORUS-AQ period. All the gas species measured in Korean stations have the same ppmv unit as in WRF-Chem, but all
the Chinese sites report the data in $\mu$g/m$^3$, requiring a unit conversion as part of observation processing. Using the molecular
weight of each gas species ($w_{gas}$) and the molar volume of a gas ($V_m$ = 22.4 L/mol) at 1 standard temperature and pressure
(STP), the unit of the Chinese data is converted as [ppmv] = [$\mu$g/m$^3$] x $V_m$ / $w_{gas}$ / 1000.

As the surface stations are mostly concentrated in the large cities, the hourly data that belong to the same 9-km model grid
are randomly split for assimilation and verification, then each dataset is averaged over each grid. As a result, 279 Korean sites
are averaged into 219 stations (or grids) for assimilation, while the other 100 sites are averaged to 71 independent observations
for evaluation over South Korea. The Chinese data has a lot of missing values, especially for the period of heavy pollution
events (24-26 May 2016), and because the verification is made over Korean sites only, they are used without such processing.

Data quality control (QC) is done by setting maximum thresholds of observation values and innovations ((o-f)'s) during
the assimilation procedure. Surface PM$_{2.5}$ and PM$_{10}$ observations are rejected when they are greater than 300, 500 $\mu$g/m$^3$,
respectively, or are different from their model equivalent (e.g., $\mathbf{H}(\mathbf{x_b})$) by more than 100 $\mu$g/m$^3$. Gas species are also checked





with the maximum threshold of 2, 2, 2, and 20 ppmv for the observed so$_2$, no$_2$, o$_3$, co, respectively. They are also rejected based on the threshold of 0.2 ppmv for the innovations.

Gas-phase pollutants on the ground are assimilated together, as opposed to individual species, using the corresponding model variables as their analysis (or control) variables. As observations for all the gas species are processed to have the same ppmv unit as the model variables before assimilation, the observation operator becomes a simple horizontal interpolation (e.g.,

bilinear interpolation) of the corresponding variable at the lowest model level. For the new assimilation capability, several new parameters are added to namelist.input in WRF-Chem, as summarized in Table 2. To demonstrate the capability of all the new observation operators (that are independent of each other), this study only presents the simultaneous assimilation of all six pollutants using chemicda_opt = 4, as listed in Table 2.

In this 3D-Var analysis, observation errors are assumed uncorrelated such that the observation error covariance matrix **R**

in Eq. (1) becomes diagonal with the observation error standard deviations as diagonal elements. For the gas species, the observation error is simply assigned as 10% of the observed value regardless of the location. For surface PM concentrations, the observation error is estimated as a sum of the measurement error ($\epsilon_o$) and the representative error ($\epsilon_r$) as $\epsilon_x = \sqrt{\epsilon_o{}^2 + \epsilon_r{}^2}$, following Elbern et al. (2007). The measurement error increases linearly with the observed value ($x_o$) as $\epsilon_o = 1.5 + 0.0075 * x_o$ while the representative error is formulated as $\epsilon_r = \gamma \epsilon_o \sqrt{\frac{\Delta x}{L}}$ where $\gamma$ is set to be 0.5, $\Delta x$ is grid spacing (here, 27 km for

domain 1 and 9 km for domain 2) and the scaling factor L is defined as 3 km, as in Ha et al. (2020).

## 3  Cycling experiments

A month-long cycling experiment is conducted, assimilating all the surface observations for six pollutants (PM$_{2.5}$, PM$_{10}$, so$_2$, no$_2$, o$_3$, and co) in both domains every 6 h. The baseline experiment ("NODA") is first conducted, recycling 6 h forecasts from the previous cycle. And the background error statistics are computed from the extended forecasts (e.g., up to 48 h) in

NODA. Then, the DA experiment assimilates all the observations to update the analysis every 6 h based on the background error covariance, using the same input data and the same lateral boundary conditions for both meteorological and chemical fields as in NODA.

The UM global forecasts are initialized from the UM analysis at 18 UTC every day so that the UM global analysis is used for 18 UTC cycles while the following 6-18 h UM forecasts are employed at 00-12 UTC cycles next day, respectively. The

UM simulations run by KMA define surface fields at 1.5 meters and the soil moisture content at 0-0.1, 0.1-0.35, 0.35-1.0, and 1-3 m soil layers in the unit of [kg/m$^2$/TS] where TS indicates the thickness of each soil layer. To ungrib the data correctly, the Vtable and the ungrib source codes in WPS are modified accordingly.

Figure 5 depicts the time series of hourly surface PM$_{2.5}$ concentrations, averaged over 71 evaluation sites in South Korea for the last week of May 2016 with heavy pollution events. Observations are marked in blue dots, and the hourly forecasts in

DA and NODA are drawn in solid red and black lines, respectively. The NODA experiment concatenates 0-6 h forecasts every cycle while DA presents the analysis every 6 h and 1-5 h forecasts for other times. The 3DVAR analysis and the subsequent





forecasts in DA follow the observations closely, but without data assimilation, 0-6 h forecasts in NODA largely deviate from the measurements, even beyond the observation uncertainty across stations (shaded in light blue).

Figures 6 and 7 illustrate observation-minus-background (omb; dotted gray line) and observation-minus-analysis (oma; solid
black line) in DA, as a time series of surface PM concentrations and gas species, respectively, for the entire month. The total number of observations (blue dots in Fig. 6) varies from cycle to cycle, but the time series of omb and oma indicates that the analysis system gets spun up quickly (with the steady trend of oma) and runs reliably throughout the month-long period with the analyses closer to observations than background forecasts for all six pollutants. The number of observations in trace gases is omitted in Fig. 7 because it is very similar to that of PM observations and the omb in gas species is greatly fluctuating
with cycles. Such large oscillations of omb and large differences between analysis and background are often attributable to the considerable errors in the forecast model and/or forcing parameters, which prompt the model state to return to its own equilibrium quickly (e.g., within 6 h). For the rest of the figures, the evaluation is made only for 7-31 May 2016, discarding the first week of cycling as a spin-up period.

The MADE-VBS aerosol parameterization has been reported to simulate the chemical composition over the East Asian
region reasonably well (Lee et al., 2020; Saide et al., 2020). As seen in Fig. 8, the surface $PM_{2.5}$ analysis is dominated by nitrates (NO3), sulfate (SO4), ammonium (NH4), unspeciated $PM_{2.5}$, and anthropogenic secondary organic aerosols (ASOA) in Seoul, South Korea (in that order). Compared to the mean analysis for the evaluation period, the analysis in the heavy pollution event on 26 May 2016 shows that major constituents' contributions further increase, particularly by nitrate. Due to the limited information content of atmospheric composition measurements as well as the scarcity of such observations, it is
hard to evaluate the fractional aerosol contribution by all the species defined in the MADE-VBS scheme. Hence, the pie charts are presented only to understand how the aerosol composition is represented in the parameterization and how it changes with high $PM_{2.5}$ concentrations. But the overall composition with major constituents seems consistent with those from previous studies (Jo et al., 2020; Tian et al., 2019). Because the forward operator assigns an equal weight for each aerosol species, the fractional contribution is not substantially changed by assimilation in this particular case study.

Figure 9 presents the horizontal distribution of analysis increments in the assimilated variables at the lowest model level, averaged for the evaluation period. Surface $PM_{2.5}$ concentrations are reduced by assimilation, especially over the middle east China (along 30°N), indicating that they were mostly overpredicted in background forecasts, likely due to the systematic overestimation of anthropogenic emission data. Given that air pollutants in the emission data constitute the majority of the precursors of $PM_{2.5}$ pollution, surface $PM_{2.5}$ concentrations could strongly depend on emissions, which might have led to the
overestimation in the background forecasts. Therefore, the assimilation of surface $PM_{2.5}$ tends to counteract the overestimation driven by the emission data over China. On the contrary, $PM_{10}$ concentrations are predominantly enhanced by assimilation over most areas, presumably because coarse mode aerosols might not be sufficiently described in both the emission data (through "E_PM_10") and the model estimate. Among the coarse-mode species, dust aerosols (soila) show the most significant analysis increments over the Jing-Jin-Ji (an abbreviation of the Chinese names of Beijing, Tianjin, and Hebei) Metropolitan region (not
shown). On the other hand, the analysis does not make meaningful changes in $so_2$ and $no_2$ that have short lifetime, but tend to decrease ozone and increase carbon monoxide over South Korea.





Figure 10 examines how the vertical distribution of aerosol species systematically changes with the assimilation over domain 2. Even if only the surface observations were assimilated, the entire boundary layer is affected by continuous cycling, based on the aerosol forecast error structure. The DA experiment mainly reduces most aerosol species contributing to $PM_{2.5}$ in the

boundary layer. But soil-derived dust and sea salt aerosols are significantly increased in the low troposphere, owing to their large standard deviations and eigenvalues estimated in the background error covariance. These coarse mode variables could be also affected by weather conditions to a greater extent as they could be more sensitive to the large-scale advection like low-level jets. The role of meteorological fields or their interactions with aerosols will be examined in the context of concurrent data assimilation of chemical and meteorological observations in the following study.

The reduction of $PM_{2.5}$ and the large increase of $PM_{10}$ in the boundary layer, as shown in Fig. 11, are consistent with previous results. $PM_{2.5}$ shows that the analysis (orange) and the following 6 h forecast (red) are not much different in the climatological sense (e.g., mean over time and space). But $PM_{10}$ displays relatively large discrepancies between the two and even bigger differences between NODA and DA, mainly due to the large changes in soila, as shown in the previous figure. Systematic disparities between observations and the model estimates typically imply the deficiencies in the model simulation and/or the

forcing parameters. As the focus of this study is the prediction of surface $PM_{2.5}$ concentrations, no further investigation is made on the systematic errors in $PM_{10}$ simulations in this study. But generally, the larger the model error gets, the harder it is to make an optimal solution in the analysis. On the other hand, sulfur dioxide and nitrogen dioxide near the surface are slightly increased by assimilation over domain 2, which does not last for 6 h because of their short lifetime. While the changes in the vertical structure of those two fields are confined to the boundary layer, ozone and carbon monoxide experience the adjustments

for the entire profile through the cycling.

Figure 12 illustrates the time series of rms and bias errors of 0-48 h forecasts with respect to independent $PM_{2.5}$ observations at the surface. The large initial errors in NODA imply that aerosol species are not properly initialized without assimilation, even if they are recycled every 6 h for the whole month. With data assimilation, initial conditions in the DA experiment are substantially improved over both domains, leading to smaller forecast errors throughout 48 h forecasts. In domain 1, a large

overestimation in NODA is significantly reduced by assimilation, but the positive bias remains for 48 h. In domain 2, the systematic bias is mostly corrected in DA up to 36h forecasts, and the rms errors are consistently small compared to NODA. The forecast errors are mostly distinguishable for the first 24 h, and the analysis impact typically lasts no longer than 6 h in trace gases like $no_2$ and $so_2$, 24 h forecast mean errors are thus summarized in Table 3 for all six pollutants. Comparing to the baseline run (NODA), the DA experiment systematically improves surface $PM_{2.5}$ forecasts in both domains, with the rms

errors decreased by 26% and 20% over domains 1 and 2, respectively. The rms errors of $PM_{10}$ are reduced only by ~14% in DA, but the systematic underestimation gets mostly diminished over both domains. The assimilation is not very effective in the prediction of gas species except for carbon monoxide, partly due to the model errors and partly due to the observation errors that might need to be further adjusted for better results.

The forecast errors depicted in Fig. 12 are dominated by moderate (or clear-sky) cases in Korea, but air quality forecasting

becomes more crucial for heavy pollution events, making the categorical forecast verification important in a practical sense. Table 4 categorizes four different events based on hourly surface concentrations in six pollutants, and Table 5 defines categorical



forecasts for different air pollution events. Figure 13 highlights the forecast accuracy for categorized events, verified against independent observations, based on the formulae described below.

$$Overall\_Accuracy(\%) = \frac{a1 + b2 + c3 + d4}{N} \times 100 \qquad (5)$$


$$High\_Pollution\_Accuracy(\%) = \frac{c3 + d4}{III + IV} \times 100 \qquad (6)$$

$$False\_Alarm(\%) = \frac{II}{II + IV} \times 100 \qquad (7)$$

where $I = a1 + a2 + b1 + b2$, $II = c1 + c2 + d1 + d2$, $III = a3 + a4 + b3 + b4$, $IV = c3 + c4 + d3 + d4$, and $N = I + II + III + IV$.

The air quality forecasting operated by the NIER in Korea is currently evaluated in the same way daily, except for daily mean values (rather than hourly averages). The forecast accuracy rates defined here can be considered as skill scores for categorized events so that the higher, the better. First of all, NODA shows very stable accuracy rates between 40 and 50% for all events. As forecast errors usually grow with time due to the model uncertainty, this means that the forcing parameters consistently constrain the chemical forecasting. With data assimilation (red), the initial accuracy gets doubled up in both domains (up to

80%), even for high pollution cases. But the benefit of the analysis quickly disappears with time, implying the challenges with chemical data assimilation using the 3DVAR technique and large uncertainties in aerosol simulations. Note that the DA algorithm used here cannot produce an optimal solution when there are significant errors in the model and/or the forcing parameters as the strong-constraint variational system assumes a perfect model in the optimization. As Bocquet et al. (2015) pointed out, even with the improved analysis, it is hard to compete with forcing parameters such as emissions by which the

chemical transport model is strongly driven, making the chemical analysis impact typically limited to the first-day forecasts. The results shown here are consistent with previous studies, illustrating that the most benefit of data assimilation is limited to the first 24h forecasts, although the overall forecast accuracy in DA still remains higher than NODA up to 48 h. In domain 2, due to the small sample size, the forecast accuracy for high concentrations is not as consistent (and smooth) as in domain 1, and the results may not be statistically significant. But the false alarm rates for all events are also clearly reduced for 0-24 h

forecasts, indicating that the assimilation systematically improves the 9-km simulations.

## 4 Conclusions and discussion

This study introduced a new extended version of the WRFDA 3DVAR system for the RACM chemistry and the MADE/VBS aerosol parameterization in WRF-Chem and demonstrated the analysis capability for surface observations for six pollutants ($PM_{2.5}$, $PM_{10}$, $so_2$, $no_2$, $o_3$, and co).

For May 2016, the RACM and MADE/VBS chemical schemes simulated nitrates, sulfate, ammonium, and anthropogenic secondary organic aerosols (ASOA) as the largest constituents of the predicted $PM_{2.5}$ over Korea. The month-long cycling



experiments confirmed that chemical assimilation could improve the forecasts mainly for 24 h, verified against independent observations. The improvements were also clear in the heavy pollution events (24-26 May 2016) over East Asia, suggesting that the new system can be useful for advancing air quality forecasting, even for predicting exceedance and non-exceedance events.

Given the lack of interoperability between chemical parameterization schemes, these results validate that the MADE/VBS scheme can improve air quality forecasting in the context of chemical weather cycling, especially over the East Asian region. The new codes developed here will be included in the next release of the WRFDA system.

Even with the successful demonstration of the new implementation, however, the 3DVAR analysis system can be further improved or examined to maximize the impact of chemical observations. In an effort to optimize the system design, con-

ventional weather data were assimilated at the same time, and chemical boundary conditions were updated every 6 h. Many processes and inputs (ex. emissions) depend on meteorological conditions, and the large-scale chemical forcing can affect the quality of background forecasts, but their roles on aerosol prediction were not examined, focusing on the demonstration of the new development. Neither were examined large uncertainties in the forecast model and the emission data, and the observation errors that may need to be tuned, especially for trace gases. Those important aspects are left behind for future studies. And

a more appropriate choice of control variables, for example, can enhance the conditioning of the 3DVAR problem (Courtier and Talagrand, 1990). The observation operator used in this study treated each aerosol species in each mode as an individual control variable, but because the Aitken mode variables contributed to the particulate matter concentrations only for about 10 %, it might be worth trying to reduce the total number of control variables by either using major constituents or combining Aitken and accumulation modes for each aerosol variable.

The error cross-correlations between meteorological variables and chemical species or between chemical species could not be specified in the current variational data assimilation framework but might also play a critical role in improving air quality forecasting, particularly for long-range transport of air pollutants that often cause heavy pollution events over Korea. For a dynamical estimation of such cross-correlations, ensemble-based methods should be introduced.

*Code and data availability.* The WRF-Chem v3.9.1 codes are freely available from https://www2.mmm.ucar.edu/wrf/users/downloads.html.

The WRFDA source codes developed for this study and the WPS codes modified for the UM grib data can be downloaded from https://doi.org/10.5281/zenodo.4594671. Chemical observations are available at http://www.airkorea.or.kr/ for Korean sites and at http://www.cnemc.cn for Chinese stations. The NCEP prepbufr data are accessible in https://rda.ucar.edu/datasets/ds337.0/. CAM-Chem model outputs for lateral boundary condition files can be downloaded from https://www.acom.ucar.edu/cam-chem/cam-chem.shtml. And the WRF-Chem preprocessor tools such as mozbc and megan_bio_emiss are available at https://www.acom.ucar.edu/wrf-chem/download.shtml.

*Author contributions.* Ha developed all the codes, designed the experiments, analyzed the results, and wrote the manuscript.

*Competing interests.* The author declares that there is no conflict of interest.



*Acknowledgements.* All the experiments presented here were performed on the Cheyenne supercomputer at the National Center for Atmospheric Research (NCAR). This research was jointly supported by the National Center for Atmospheric Research which is a major facility sponsored by the National Science Foundation under Cooperative Agreement No. 1852977, and a grant from the National Institute of Environment Research (NIER), funded by the Ministry of Environment (MOE) of the Republic of Korea (NIER-2019-01-02-087). The author acknowledges the use of the WRF-Chem preprocessor tool (mozbc and megan_bio_emiss) provided by the Atmospheric Chemistry Observations and Modeling Lab (ACOM) of NCAR. The UM grib files were read by kwgrib2, provided by Yonghee Lee at the Korean National Institute of Environmental Research (NIER).



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




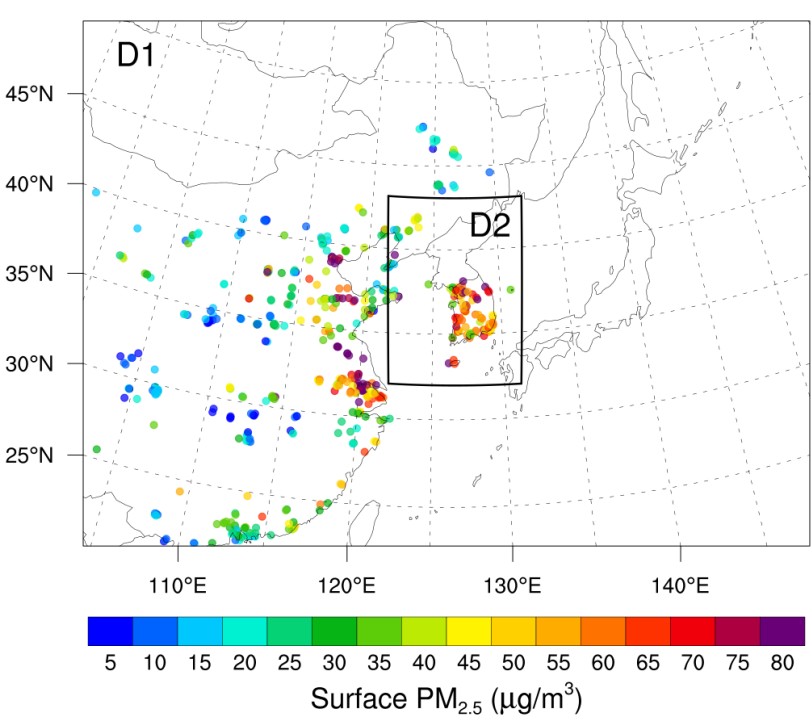

**Figure 1.** The surface observation network in two model domains. A black box indicates domain 2 (D2) over the Korean peninsula, nested down from domain 1 (D1). Colored dots indicate surface PM$_{2.5}$ observations assimilated at 2016-05-26_00:00:00 UTC.





**Figure 2.** Vertical profile of background error standard deviations for aerosol species in a) accumulation b) Aitken and c) Coarse modes, and d) gas species over domain 2.





**Figure 3.** Vertical auto-correlations in four major aerosol species in accumulation and Aitken modes (in the first and second rows, respectively), three coarse mode aerosols (in the third row), and four gas species (in the bottom row) over domain 2, contouring from 0.1 to 0.9 every 0.1.





**Figure 4.** The horizontal length scales of the same species as in Fig.3.



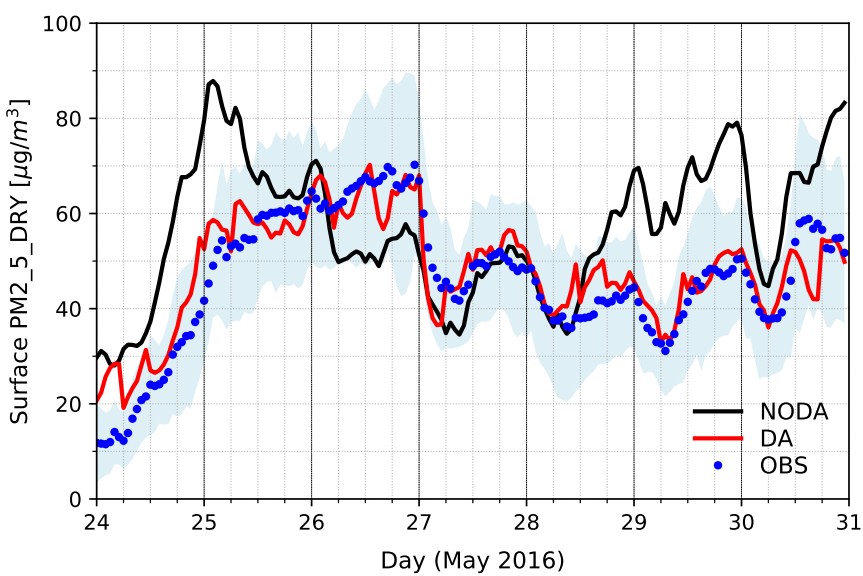

**Figure 5.** Time series of hourly PM$_{2.5}$ concentration on the ground. The baseline experiment ("NODA") is plotted in black, while the DA experiment with the analysis every 6 h in red. Observations averaged over all the evaluated stations in South Korea are marked as blue dots, enclosed with the shaded area in light blue for the standard deviation in observations.



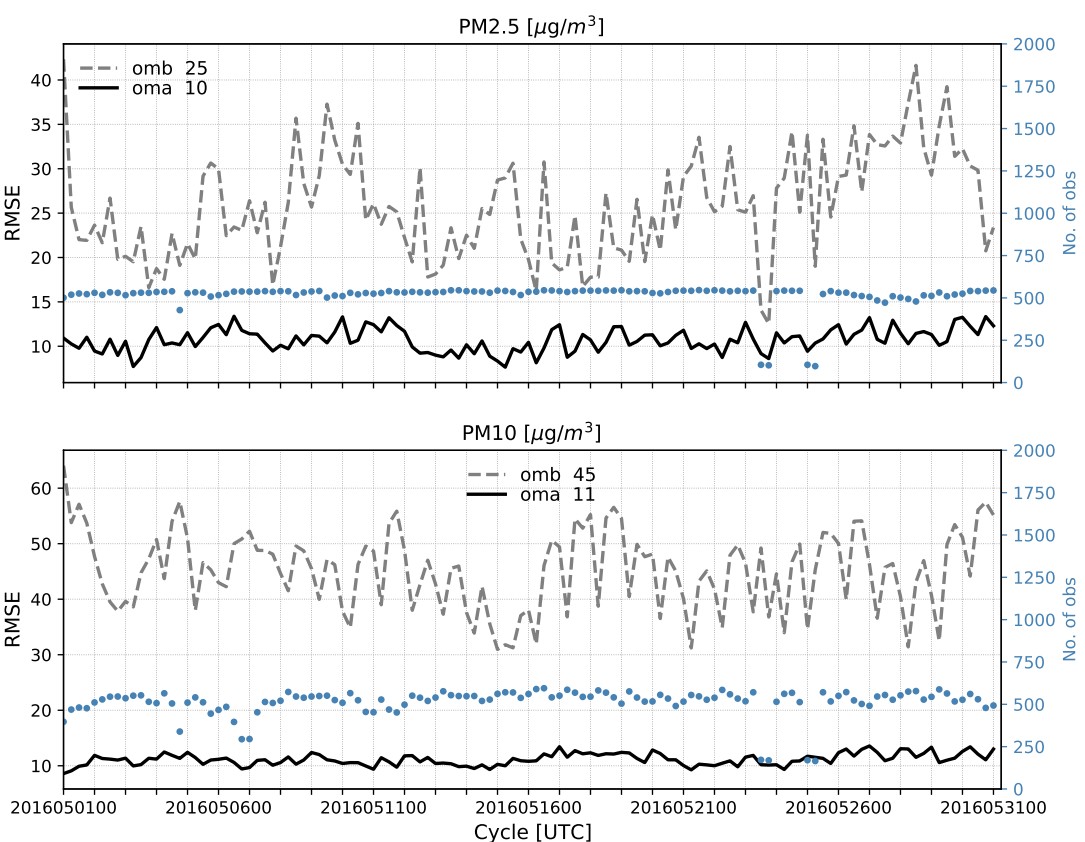

**Figure 6.** Time series of (o-a)'s and (o-b)'s in surface $PM_{2.5}$ (top) and $PM_{10}$ (bottom) in the "DA" experiment, as root-mean-square error over all the stations assimilated in domain 1 at each cycle. The averages over cycles are shown in the legend, and the total number of stations at each cycle is marked in blue dots with the y-axis on the right.





**Figure 7.** Same as Fig. 6, but for gas species.



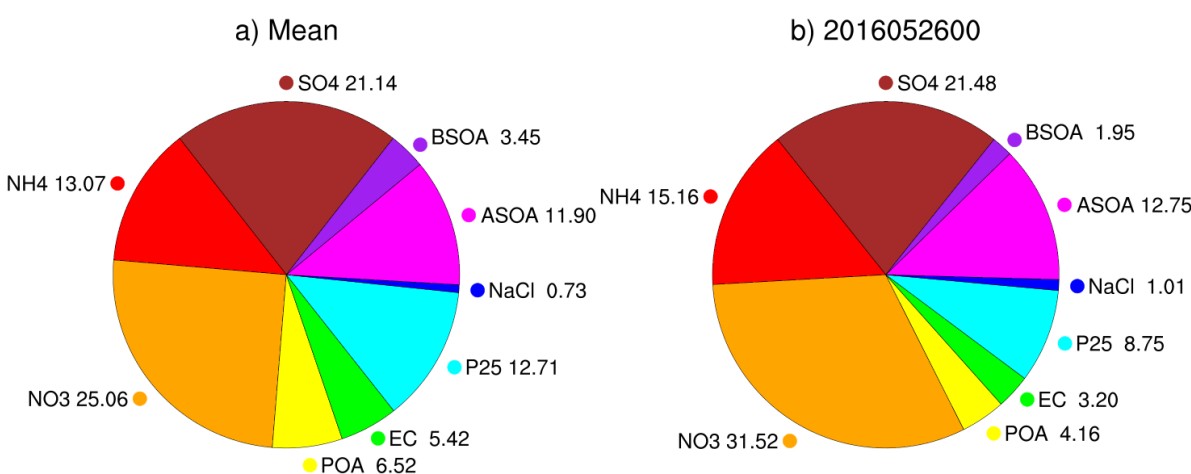

**Figure 8.** Pie charts showing the percentage contribution by aerosol species in Seoul, South Korea. a) The analysis averaged for 97 cycles from 00 UTC May 7 to 00 UTC 31 May 2016 and b) the analysis at 2016-05-26_00:00:00 UTC over domain 2 in the DA experiment. Surface $PM_{2.5}$ consists of aerosol sulfate (SO4; so4ai and so4aj), ammonium (NH4; nh4ai and nh4aj), nitrate (NO3; no3ai and no3aj), primary organic matter (POA; orgpai and orgpaj), elemental carbon (EC; eci and ecj), unspeciated $PM_{2.5}$ (P25; p25ai and p25aj), sodium chloride (NaCl; naai, naaj, clai, and claj), 4-bin anthropogenic and biogenic secondary organic aerosols (ASOA and BSOA, respectively) at the lowest model level.



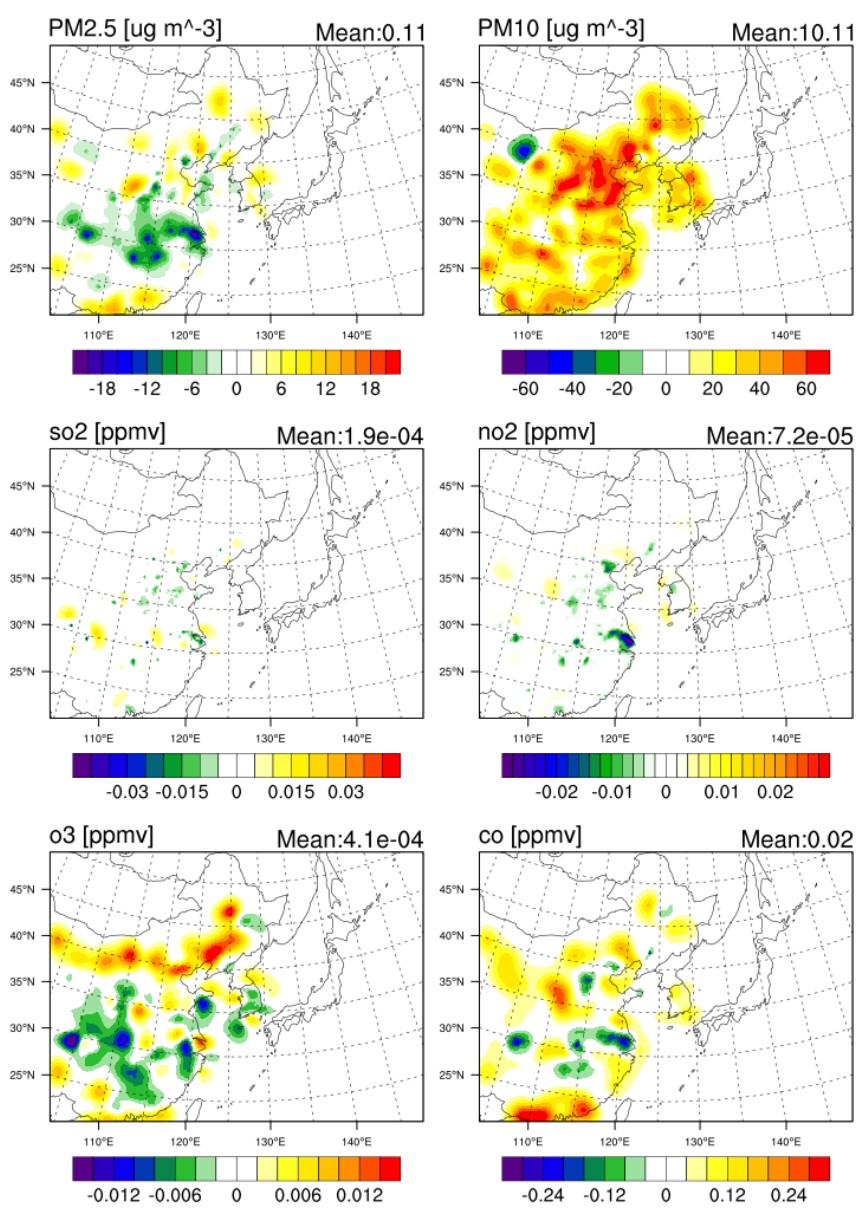

**Figure 9.** Horizontal distribution of the analysis increments in particulate matter concentrations and four gas species at the lowest model level over domain 1 in the DA experiment, averaged for 7 - 31 May 2016. The domain average is shown in the top right corner of each panel.





**Figure 10.** Vertical profile of aerosol species ([μg/kg-dryair]) averaged over domain 2. The 6 h forecasts in NODA (black) and DA (red) are depicted in solid lines, while the analyses in DA in the dotted orange line.



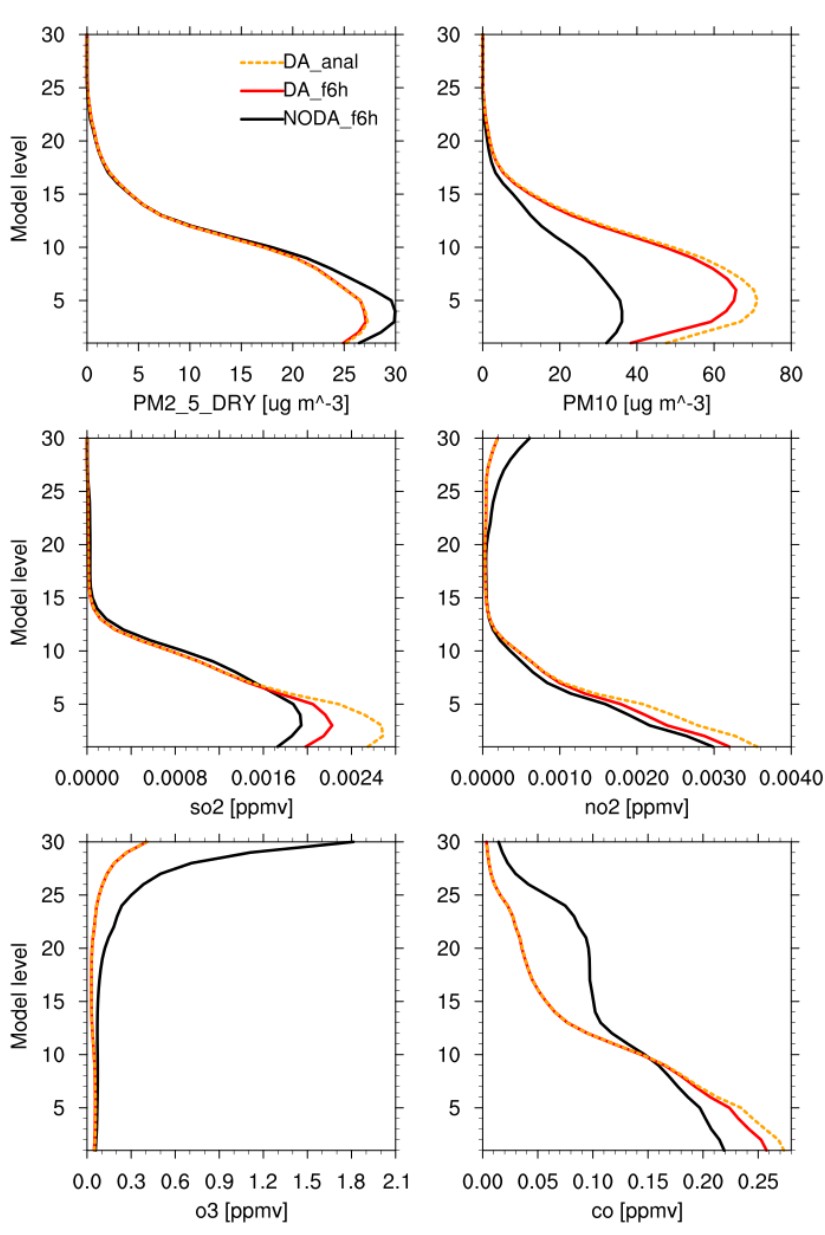

**Figure 11.** Same as Fig. 10, but for six pollutants assimilated in DA.



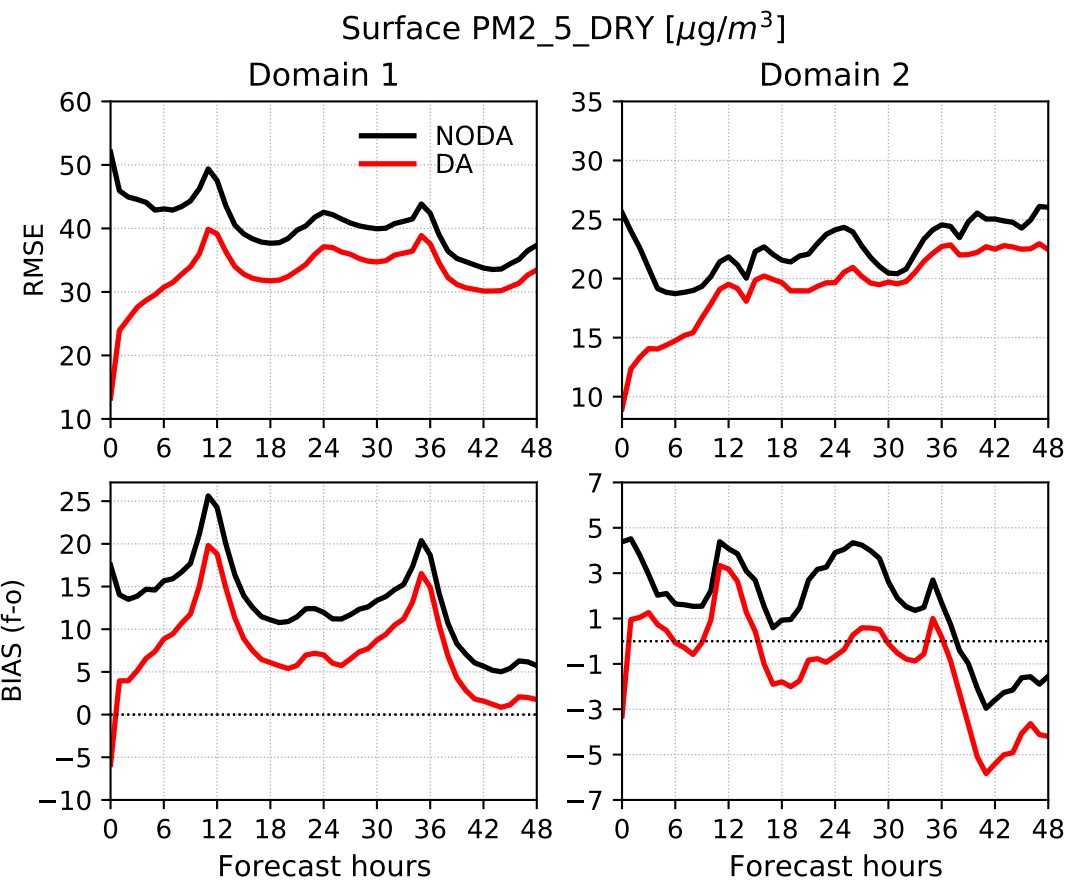

**Figure 12.** Time series of (top) root-mean-square-error (rmse) and (bottom) bias as (forecast-minus-observation) in surface $PM_{2.5}$ concentration of the hourly forecasts from the analysis at 00 UTC from 7 to 31 May 2016. The left panel shows the errors of forecasts at 27-km resolution verified against 1188 stations over domain 1 and the right panel presents the 9-km forecast errors with respect to surface $PM_{2.5}$ observations from 71 independent stations in South Korea.



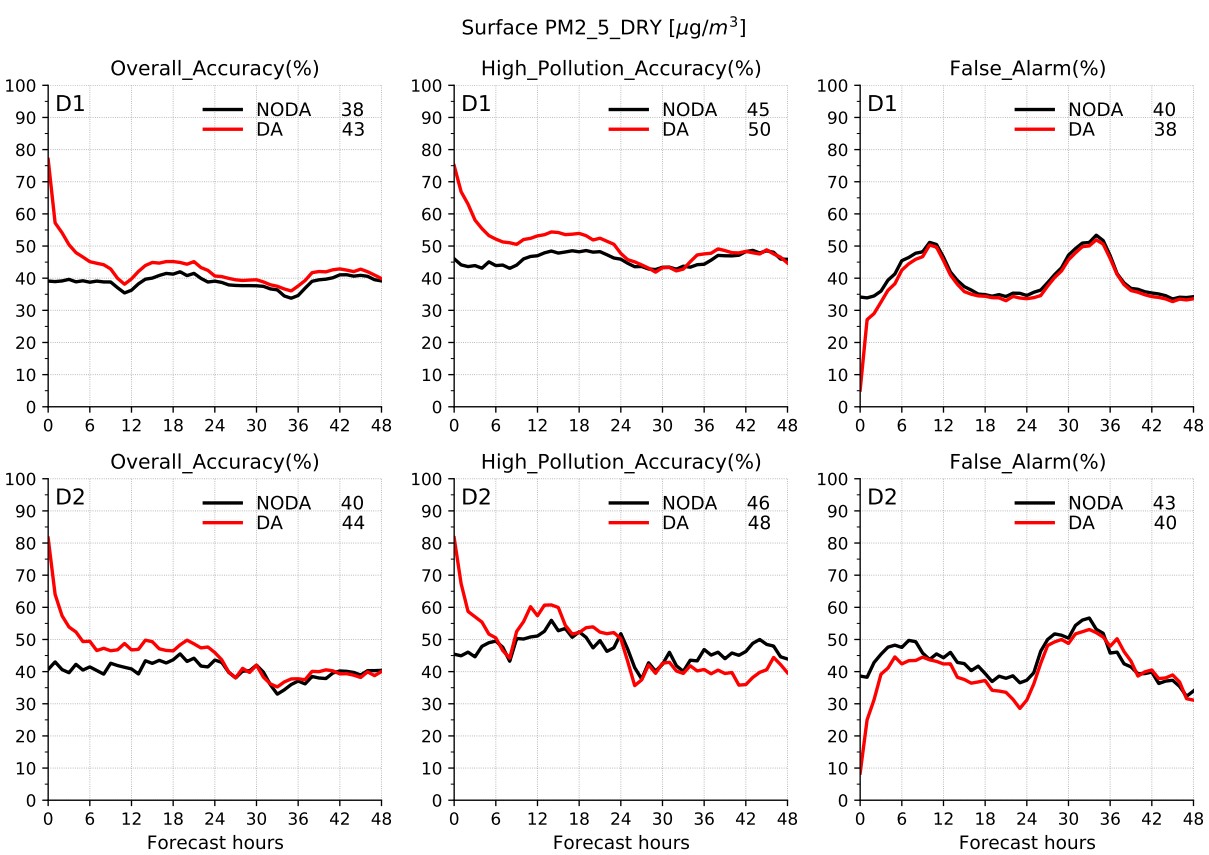

**Figure 13.** Same as Fig. 12, but for the forecast accuracy (%) for categorized events, as defined in Tables 4 and 5.





**Table 1.** Summary of WRF-Chem physics configuration.

| Physical processes | Parameterization schemes |
|---|---|
| Aerosol chemistry | RACM |
| Gas-phase chemistry | MADE-VBS |
| Photolysis | Madronich |
| Cloud microphysics | Lin |
| Cumulus | Grell 3D ensemble |
| Longwave radiation | RRTMG |
| Shortwave radiation | RRTMG |
| PBL | YSU |
| Surface layer | Monin-Obukhov |
| Land surface | Noah |

**Table 2.** A list of new namelist parameters in WRFDA.

| | namelist options | description |
|---|---|---|
| &wrfvar7 | chem_cv_options = 108 | racm_soa_vbs_kpp |
| &wrfvarchem | use_chemic_surfobs = .true. | chemical DA |
| &chem | chemicda_opt = 1 | 1 = pm2.5 |
| | | 2 = pm10 |
| | | 3 = pm2.5 and pm10 |
| | | 4 = all (pm2.5, pm10, so2, no2, o3, co) |
| | | 5 = gas only (so2, no2, o3, co) |




**Table 3.** The rmse and bias errors of 24 h forecasts in NODA and DA experiments.

| | | rmse | | bias | |
|---|---|---|---|---|---|
| | | D1 | D2 | D1 | D2 |
| $PM_{2.5}$ | NODA | 42.8 | 21.3 | 15.4 | 2.5 |
| | DA | 31.6 | 17.0 | 8.4 | 0.0 |
| $PM_{10}$ | NODA | 53.4 | 33.0 | -9.6 | -13.9 |
| | DA | 46.1 | 28.2 | -0.7 | -1.9 |
| $so_2$ | NODA | 0.016 | 0.006 | 0.007 | 0.001 |
| | DA | 0.015 | 0.006 | 0.007 | 0.001 |
| $no_2$ | NODA | 0.018 | 0.001 | 0.017 | -0.001 |
| | DA | 0.017 | 0.000 | 0.017 | -0.003 |
| $o_3$ | NODA | 0.023 | 0.007 | 0.02 | 0.006 |
| | DA | 0.022 | -0.006 | 0.02 | -0.004 |
| $co$ | NODA | 0.662 | -0.209 | 0.277 | -0.228 |
| | DA | 0.642 | -0.182 | 0.249 | -0.184 |

**Table 4.** Air quality index values, as defined in the NIER in Korea.

| Concentration (hourly) | Good | Moderate | Unhealthy | Very Unhealthy |
|---|---|---|---|---|
| $PM_{2.5}$ [$\mu g/m^3$] | 15 | 35 | 75 | > 75 |
| $PM_{10}$ [$\mu g/m^3$] | 30 | 80 | 150 | > 150 |
| $so_2$ [ppmv] | 0.02 | 0.05 | 0.15 | > 0.15 |
| $no_2$ [ppmv] | 0.03 | 0.06 | 0.20 | > 0.20 |
| $o_3$ [ppmv] | 0.03 | 0.09 | 0.15 | > 0.15 |
| $co$ [ppmv] | 2 | 9 | 15 | > 15 |

**Table 5.** Categorical forecasts for different air pollution events.

| Category | | Forecast | | | |
|---|---|---|---|---|---|
| | | Good | Moderate | Unhealthy | Very Unhealthy |
| Observation | Good | a1 | b1 | c1 | d1 |
| | Moderate | a2 | b2 | c2 | d2 |
| | Unhealthy | a3 | b3 | c3 | d3 |
| | Very Unhealthy | a4 | b4 | c4 | d4 |