# Peer review of "Implementation of aerosol data assimilation in WRFDA (V4.0.3) for WRF-Chem (V3.9.1) using the MADE/VBS scheme"

_Geoscientific Model Development, 2021_

## Author Comment (AC1)

RC1: 'Comment on gmd-2021-74', Anonymous Referee #1, 27 Jul 2021  reply

Thank you for the review of this manuscript. Based on your comments, Figure 8 is now replotted and all the corrections were made as suggested. My point-by-point responses are found below.

*Reviewer's comments are italicized.

*The present work of Ha demonstrates an interface between the WRF-Chem model and the WRFDA system. Said interface is used for surface assimilation of PM2.5, PM10 and four gas species (SO2, NO2, O3 and CO). A comparison against independent surface observations recorded over the Korean Peninsula indicates that 3DVAR PM2.5 forecasts produced by the WRFDA system have a lower RMSE than that of a non-DA baseline, although this does not appear to be true for all chemical species under consideration. 3DVAR increases the overall accuracy of PM2.5 forecasts (for both time series and categorized events, see figures 12 and 13). However, the experiments do not appear to show a statistical significant improvement in the "false alarm" rates over those of the non-DA baseline in either of the two model domains (cf. right panels in Figure 13). Nevertheless, the manuscript is well written and the work of Ha should facilitate further developments on top of the existing WRFDA implementation (as described in the "Conclusions" section). I believe it warrants publication in GMD after minor revisions, as follows:*

*1/ Abstract: "co" -> "CO".*

=> "co" is now changed to "CO".

*"And the effects" -> "The effects".*

=> "And the effects" is also changed to "The effects".

*Also I feel like the abstract is too "optimistic" re. the improvements in forecast skill over the non-DA baseline (particularly the final sentence). This should be qualified to be consistent with the actual results presented in sections 3 - 4.*

=> Based on the results from Table 3, the final sentence is now modified as below.

", reducing systematic bias errors in surface PM2.5 (PM10) concentrations to 0.0 (-1.9) µg/m$^3$ over South Korea in 24-h forecasts."

*2/ l. 60 (p. 3): "readers refer to" -> "readers are referred to"*

=> Changed.

*3/ l. 139 (p.5): "so2, no3, o3 and co" - > should be all capitalized (there are other instances of inconsistent capitalization throughout the manuscript)*

=> The four gas species are now capitalized everywhere in the text: L6, L149, L191, L218-220, L256, L260, L275, L291-292, L405-406. Also, they are now capitalized in Table 2.

*4/ l. 231 (p. 8): define "(o-f)'s" as "observations-minus-forecasts". this should be consistent with the labels in e.g. Figure 6 ("omb", "oma", ...)*

=> The paragraph is now removed in the process of polishing the manuscript. But the caption of Figure 6 is now modified from (o-a)'s and (o-b)'s to observation-minus-background (omb; dotted gray line) and observation-minus-analysis (oma; solid black line).

*5/ Figure 8: The differences between the averaged analysis ("Mean") and the May 26 analysis is difficult to interpret, I suggest using a different layout to display the information (_not_ a pie chart).*

=> Figure 8 is now replotted to show the differences in a bar plot in b). The caption is also edited accordingly.

[Figure]

**Figure 8.** a) A pie chart showing the percentage contribution by aerosol species in Seoul, South Korea, in the analysis averaged for 97 cycles from 00 UTC 7 May to 00 UTC 31 May 2016 and b) deviations from the mean analysis in the analysis at 00 UTC 26 May 2016 over domain 2 in the DA experiment. Surface $PM_{2.5}$ consists of aerosol sulfate (SO4; so4ai and so4aj), ammonium (NH4; nh4ai and nh4aj), nitrate (NO3; no3ai and no3aj), primary organic matter (POA; orgpai and orgpaj), elemental carbon (EC; eci and ecj), unspeciated $PM_{2.5}$ (P25; p25ai and p25aj), sodium chloride (NaCl; naai, naaj, clai, and claj), 4-bin anthropogenic and biogenic secondary organic aerosols (ASOA and BSOA, respectively) at the lowest model level.

*6/ l. 379 - 380: The "significant reduction" in false alarm rates is not evident to this reviewer from the results presented, particularly for domain D1 (Fig. 13).*

=> Just for clarification, it was stated as "clearly reduced", not "significantly reduced". Respecting the reviewer's concern on the impression of being too optimistic, however, "clearly" is now removed. Also, the 9-km simulations (as mentioned in the sentence) correspond to domain 2, not domain 1. Now, "(in D2)" is added at the end, for clarification.

---

## Author Comment (AC2)

RC2: 'Comment on gmd-2021-74', Anonymous Referee #2, 15 Sep 2021

*Reviewer's comments are italicized.*

*General comment*
*This paper presents the implementation of a new interface for assimilating ground-based aerosol and gas species observations in the WRFDA/ WRF-CHEM system. The impacts of assimilating aerosol and 4 gas species observation on forecast performances is evaluated exploiting the Korea- United States Air Quality campaign.*

*While this paper is relevant for the atmospheric composition data assimilation community, it is difficult to read it and to identify the novelties. The paper has sufficient scientific content for publication, but its structure needs to be improved and the result analysis needs further developments.*

Response:

I agree with the reviewer that it may not be easy to follow all the details unless one is familiar with such a complicated cycling system for the atmospheric composition data assimilation with real observations. However, this manuscript was submitted to the "Development and technical papers" category, strictly following the guidelines provided by this journal. Development of a new interface involved numerous code changes for almost every component of the WRFDA 3DVAR system for the RACM/MADE-VBS chemistry parameterization in the WRF-Chem model. My purpose of this draft is to document the new technical development and demonstrate that it works. Through a real case study, as opposed to an idealized or a single observation test case, it was convincing that the new system performed well, even for real applications. As a technical paper, this manuscript was structured with an emphasis on the description of the new interface in Section 2, with which it can provide useful guidance for the system when it is publicly released. This work for developing a new community system is a major project with its own novelty, and is supposed to be reviewed from that perspective, I believe. A study for the scientific analysis will constitute a separate paper.

Based on the reviewer's comments, the manuscript is now extensively upgraded and polished, adding several new paragraphs to improve the clarification of the main features and three more references. Figures 2-4 are updated with the same y-axis label as the "Model level" and Table 1 now includes references, following the reviewer's suggestions. Point-by-point responses are found below.

*Main concerns*

*1. The objective of the paper is not clear. The title mentions aerosol data assimilation while in the abstract the objective of the paper seems to include both aerosol and gas species.*

=> The objective of the paper is to develop a new system for aerosol data assimilation using the RACM gas-chemistry and MADE-VBS aerosol-chemistry parameterization (chem_opt=108 in WRF-Chem), as stated in the title. The capability of assimilating gas species is included for completeness, but that is not a main development challenge because such a single variable analysis does not require any complex system development, especially for tangent linear and adjoint codes. So, the title represents the highlight of the

development work. But because this paper describes the new capabilities implemented in the WRFDA/WRF-Chem system, the abstract should include the assimilation of gas species, as part of the new features. In respect to the reviewer's point, however, the title now includes "RACM" (because it is part of the chemical option), the second last paragraph of the Introduction section is changed to "The main goal of the new system development" (by adding "main"), and one more paragraph is added as the last paragraph in Section 2.2 in the revised manuscript.

*2. The text is frequently overwhelmed by technical details and it is difficult to understand the underlying scientific message. For example, in Introduction (line 26-34) and Section2 too many configuration details are given without providing the main characteristics of the models.*

=> Please note that this manuscript is meant for a *technical* paper, belonging to the category of "Development and technical papers", as listed in https://www.geoscientific-model-development.net/about/manuscript_types.html. As such, the manuscript is supposed to provide all the technical details of the new interface. For the new development, dozens of codes with thousands of lines were developed in WRFDA, expanding the current structure/frame of the system. This is a significant development task, for which all the major components were modified or added. They should all be described here for the purpose of documentation. Following the guidelines provided in the link, this manuscript presents (and verifies) all the development work, which should be acknowledged within the technical development category. The main characteristics of the system were already described in Section 2, but only from the technical aspects, as expected. Based on the reviewer's comment, however, the paragraph in L26-34 is entirely rewritten. See my response to your specific comment #2, below.

*3. The structure of the text needs substantial improvement to facilitate the reading (and the review) of the text. A dedicated result section is missing. Some general statements on data assimilation are given in Section2 while they should be moved to Introduction or removed. Section 3 is mixing methodology details, results and discussions. The results should distinguish i) those related to the "technical evaluation" of the assimilation (e.g first guess departure) and ii)those related to the interpretation of the analysis and forecast fields in relation with transport, emissions, process representations …*

=> In respect to the reviewer's comment, the title of Section 3 is now changed to "Results from cycling experiments". This section is mainly to demonstrate that the new interface was successfully implemented, for which experiments and results are combined together, on purpose. Along the line, section 2 is supposed to be more informative than section 3, and all the results presented in section 3 are basically to confirm that the new system works well, through the "technical evaluation". Hopefully, the new system development can pave the way towards more scientific findings, but a more in-depth analysis of the current results for scientific messages is beyond the scope of this study. On the other hand, Section 1 was dedicated to the introduction of aerosol data assimilation and the motivation of the new development, not meant for the general description of data assimilation. Structure-wise, no changes are made.

*4. The analysis of the results is sometimes too vague and limited to very well-known facts. This work potentially conveys very interesting scientific results which are not reflected in the paper. The description of the results given by each Figure needs further development. I strongly recommend improving the quality of the result description using concise sentences and by removing unnecessary statements (e.g. line 315-316)*

=> Unclear exactly which part is vague unless the lines are specified. L315-316 ("the assimilation of surface PM2.5 tends to counteract the overestimation driven by the emission data over China.") are necessary in a sense that the systematic behavior of the new analysis is presented with the author's interpretation. To support the statement, though, one reference paper (Chen et al. 2019) is now added before the lines.

*5. The discussion needs further developments with reference to past studies*

=> References for the main points were already provided, but three more references for the future research are now cited at the end of the last paragraph:

"For a dynamical estimation of such cross-correlations, ensemble-based methods should be introduced (Bocquet et al., 2015; Miyazaki et al., 2020; Sandu and Chai, 2011)."

For that, two papers are now added in the References:

Miyazaki, K., Bowman, K. W., Yumimoto, K., Walker, T., and Sudo, K.: Evaluation of a multi-model, multi-constituent assimilation framework for tropospheric chemical reanalysis, Atmospheric Chemistry and Physics, 20, 931–967, https://doi.org/10.5194/acp-20-931-2020, 2020.

Sandu, A. and Chai, T.: Chemical Data Assimilation - An Overview, Atmosphere, 2, 426–463, https://doi.org/10.3390/atmos2030426, 2011.

*6. Too little background is given in Introduction to understand the novelty of this work. What is the actual contribution of this work relative to previous aerosol data assimilation studies using WRF-CHEM ?*

=> The novelty of this work is a new system development for chem_opt = 108 (e.g., the RACM gas and the MADE-VBS aerosol parameterization) in WRF-Chem/WRFDA. To provide more background of the new development, the second paragraph of page 2 in Introduction is now entirely rewritten. See my response to your specific comment #2, below.

*7. authorship: I am surprised to see a single author. Can you justify it ?*

=> As stated in the ethics guidelines in GMD, all authors listed on a presented scientific work must have contributed a significant part to it. In this study, I developed all the codes, designed all the experiments, analyzed all the results, and wrote the entire manuscript all by myself, as stated in the "Authors contributions" section.

*8. The number of Figures is too large with respect to the scientific content of the paper. I suggest selecting the most essential/informative figures*

=> This paper is not meant to deliver any scientific messages, but to prove that the new analysis system works well in full aspects. Figure 1 depicts the observing network used in the study, Figs. 2-4 describe the error characteristics in the new background error covariance (that are critical for the analysis quality), Figure 5 shows the case used in the evaluation, Figs.6-7 check the analysis performance during the cycling, Fig.8 indicate the composition of aerosol species (because PM concentrations were assimilated only as a sum of them), Figs. 9-11 the spatial distribution of analysis increments in different species, Figs.12-13 the

performance of the following forecasts (to verify that the new analysis actually led to forecast improvements). Just like the new system involves a great deal of code development in almost every part of the system, each and every component that was newly developed should be checked or evaluated. Thus, all of these figures are essential to ensure the reliability and the quality of the new system. No changes.

*Specific comments*

*1. Introduction, first paragraph: while meteorology and transport models are important sources of uncertainties, emissions, and process representation/parametrization (e.g. interactions between chemistry and aerosols, aerosol aging ...) can substantially contribute to the forecast errors.*

=> Agree, but the first paragraph is designed to explain the necessity of using the online-coupled WRF-Chem model for air quality forecasting, not to list all the factors contributing to forecast errors. But in respect to the reviewer's viewpoint, the emission aspect is now added in the new line 41 (as underlined) in "Other challenges for chemical data assimilation are large model uncertainties due to the complexity of chemical processes, significant uncertainties in forcing parameters such as emissions, highly nonlinear and non-Gaussian error distribution of chemical species, ~".

*2. Introduction, third paragraph (line 26-34): This paragraph is unclear. What is the message ?*

=> The entire paragraph is now rewritten as below. The message is that aerosol data assimilation requires a new code development for each chemistry parameterization in the WRFDA system because particulate matter (PM) concentrations consist of different sets of aerosol species depending on the chemistry option chosen from the WRF-Chem model.

"Unlike meteorological data assimilation, where most prognostic variables (except hydrometeors) remain the same regardless of the physics schemes in use, aerosol data assimilation is tied to the chemistry parameterization employed in the chemical transport model. That is because each chemical option (for both gas and aerosol chemistry) defines an entirely different set of chemical and aerosol prognostic variables. It implies that PM concentrations consist of different aerosol species, depending on the chemical option used in the WRF-Chem model. Therefore, to assimilate PM measurements, a new interface has to be developed in WRFDA for the particular chemistry option (*chem_opt*), containing new observation operators (that compute the model correspondents from the specific aerosol species), their tangent linear and adjoint models as well as the background error covariance estimation. In other words, even if the WRF-Chem model supports numerous chemical parameterization options, they cannot be used interchangeably in the WRFDA system because the analysis variables are tied to the aerosol species defined in the chemistry scheme. In practice, users should use the same chemical option between the forecast model and the analysis system, meaning that the particular chemical option should be implemented in the assimilation system in advance. That is why chemical or aerosol data assimilation studies have used a minimal set of chemical options so far and why it is challenging to use advanced chemistry in aerosol data assimilation studies within the variational analysis framework."

*3. It is difficult to understand the links between the distinct components (WRFDA, WRF-CHEM, RACM and MADE). My understanding is that MADE and RACM are 2 components of WRF-CHEM ?*

=> WRFDA produces initial conditions through the 3DVAR analysis. If it gets coupled to the WRF-Chem model through a new interface like the one presented in this study, WRFDA can provide chemical initial conditions (as well as meteorological initial conditions) to initialize WRF-Chem simulations for air quality forecasting. The WRF-Chem model has numerous chemical options combining various gas- and aerosol-chemistry parameterizations in different ways. But to be used in data assimilation, we should use the same chemical option in both WRFDA and WRF-Chem, and need to develop a new interface for whatever chemical option we choose because both systems should share the model prognostic variables to affect the aerosol prediction. In this study, the RACM gas and MADE-VBS aerosol scheme (for chem_opt = 108 in WRF-Chem/WRFDA) is chosen for such a new development. Here, MADE-VBS and RACM are bound together for the particular chemical option, although aerosol and gas chemistry parameterizations are two different components under the hood. For clarity, the first paragraph of Section 4 is now changed as "This study introduced a new extended version of the WRFDA 3DVAR analysis system for the RACM chemistry and the MADE-VBS aerosol parameterization (chem_opt = 108) in the WRF-Chem forecast model ~" by adding the underlined words. Note that the chemical option is described as the RACM/MADE-VBS scheme throughout the manuscript.

*4. Data and field campaign: A brief description of the field campaign and the associated data is missing. Giving the references is not sufficient, a brief summary should be included in this paper.*

=> A new sentence is now added in lines 65-67: "During this early summertime, air quality was measured in various platforms over the Korean peninsula and its surroundings, and long-range transport of air pollutants resulted in haze development over Korea for 25-31 May 2016.".

*5. Section 2: This section contains too general sentences (e.g. line 73-77 on the goal of data assimilation) that should be moved to Introduction or removed. This section should focus on the presentation of the WRF-CHEM modelling system.*

=> First of all, lines 73-77 (80-85 in the new version) have nothing to do with the goal of data assimilation. The paragraph is simply part of the concept of *cycling* (that was introduced in the following lines), through which the analysis and the forecast systems work together. As this section 2 is titled "The WRF-Chem analysis and forecasting system", it should present how the analysis and forecasting system works together as a whole, not just the modeling system alone. And because cycling is the way the analysis can systematically contribute to the prediction of air quality, it warrants the explanation before getting into the details of each component. But respecting the reviewer's comment, the paragraph is now polished, with the addition of "the analysis (or data assimilation) should be conducted repeatedly, ~" in L82-83 and one more sentence at the end: "For such a unified, synthetic cycling system, the analysis and the forecast systems communicate through an interface that converts between observed variables, analysis (or control) variables, and model prognostic variables."

*6. Section2: The main model characteristics (atmospheric transport, aerosol scheme, chemistry scheme) are missing.*

=> Disagree. The atmospheric transport was mentioned in lines 78-79 (as the new line numbers) in Section 2, aerosol and gas chemistry schemes were introduced in detail in the third paragraph of Section 2.1. As the references for each system/component were provided and no changes were made in the modeling system, these would suffice for this technical paper of the WRFDA development. No changes were made.

*7. Section 2.1: This section is mixing model configuration and model description. General statements on model processes, variables, forcing data sets (emission) should be included in a separate subsection.*

=> Disagree. In Section 2.1 titled "The WRF-Chem configuration", only the model description specific to the configuration was presented. No changes.

*8. Section 2.2: the general methodological statements on data assimilation (e.g role of variational assimilation lines 138-144, state vector line 153-156, Equation 1, line 195…) are sufficiently known background and it is not necessary to include them in this paper. I suggest to re-write this section by removing all general statements and replace them by appropriate references, and focus on the specific aspects of the data assimilation in WRFDA/WRF-CHEM.*

=> To introduce the new interface in the variational system, it is relevant to state what the variational analysis is and does in a single paragraph (lines 138-144, which is L148-154 in the new version) with basic equations. Also, the state vector was actually extended to accommodate chemical variables in the newly developed system; it is thus appropriate to explain the state vector in the control variables (lines 153-156 in the old version). For further clarification, however, section 2.2. is now more polished, moving Eq. (3) right next to Eq. (2) to point out why the square root of the B matrix ($B^{1/2}$) is important, This is associated with the state vector described in L153-156.

And it is essential to explain the new background error estimate as it is a critical component of the new system development, affecting the quality of the new analysis. Now, new lines are added in L200-201 to emphasize its role in aerosol data assimilation.

Finally, appropriate references were already presented for each component (Lorenc, 1986 for the 3DVAR analysis, Courtier et al. 1994 for the incremental analysis, Descombes et al. 2015 for GEN_BE). No changes.

*9. Section 2.2 line 139: This is linked to one of the general comment about the objective of the paper: is it assimilation of aerosol only or both aerosol and gas species ?*

=> As listed in Table 2, there are five different options available for what to assimilate in this new system. As stated in the first paragraph of Section 3, this study assimilated both aerosol and gas species. And aerosol data assimilation is part of chemical data assimilation, and aerosol species are considered as chemical species, in a broad sense. See my response to the main concern #1. But for clarification, "(e.g., chemicda_opt = 4)" is now added in L292.

*10. Section 2: It is not clear how the speciation of aerosol is performed while assimilating PM observations ?*

=> That was already explained with Figure 2 (now, in lines 210-212 in the new version): "During the analysis procedure, the error is used to weigh the analysis increment for a given variable, affecting how much the observed information will change the model variable."

But for further clarification, new statements are added in several places:

- L200-201: "In aerosol data assimilation, it is of particular importance as the atmospheric constituents are adjusted according to their background errors (e.g., $B^{1/2}$ in $\delta x = B^{1/2}v$)."

- L321: "~, consistent with the background error estimates shown in Fig. 2."
- L327-328: In the WRFDA system, several tuning parameters (such as var_scaling and len_scaling) are supported for further adjustments of each aerosol species, as needed.
- L408-410: "In the assimilation of surface PM concentrations, each aerosol species is adjusted according to its background errors, contributing to the atmospheric composition differently."

*11. Section 3, line 299-309 and lines 315-320: These paragraphs belong to the result interpretation or discussion. This confirms the need for a better structure of the paper to facilitate the reading.*

=> Again, Section 3 presents the results from cycling experiments, with some interpretation, to examine how the new system works. No changes.

*12. Section 3 : Figure9/line 310-320: the analysis of the correction carried out by the assimilation needs further developments. Regarding gas species, is there **any relationship between the increments for ozone and NO2** ? Regarding the uncertainties in PM background: emission can be a source of uncertainties as discussed in the text, but this can also be due to model errors. Can you comment on this.*

=> Figure 9 simply shows how the updates from the analysis look like. It would be nice to do more thorough diagnostics on why it behaves that way, but such an analysis is beyond the scope of this technical paper.

In regards to the relationship between ozone and NO2 in the analysis increments, no cross-correlations are considered between chemical species in the background error estimate, as stated throughout the manuscript (the last paragraph of the Conclusion section, for example). Thus, the relationship between the two species, even if they exist, would come from the model, not from the analysis *per se*. But to clarify this, L182 now includes "the model states (x) are directly used as control variables (v) with univariate error covariances", and L194 has "no cross-correlations are considered between the variables".

In terms of the PM uncertainties, I agree that model errors could contribute to the forecast imperfection. To what extent it contributes to the forecast error and why it works in such a peculiar way only over the middle east China, however, is another question. To be comprehensive, errors in chemical boundary conditions and soil states underneath the ground might also need to be considered in the background uncertainties. These are all good topics, but not directly related to the goal/topic of this paper. No changes.

*13. Section 3 lines 341-353: Can you discuss the differences in forecast errors as a function of forecast time between chemical species ? How does this relate to the lifetime of the chemical species and uncertainties in emission data set.*

=> Again, this paper focuses on the new assimilation capability for PM concentrations. Neither the forecast performance in gas species nor the emission uncertainty is directly associated with the main points of this study. No changes were made to stay focused on the topic.

*14. Equations 5,6,7 should be moved to method section*

=> Disagree. There is no separate method section in this manuscript. Eqs. 5-7 described how the forecast accuracy was computed for Figure 13, so they should stay in the text for the figure. No changes.

*15. Section 4: the discussion could be more developed, maybe by moving some parts of section 3 to section 4. Particularly, ==the uncertainties in emission data set and the impact of assimilation depending on the lifetime of the chemical species== are poorly discussed. A possible improvement for atmospheric composition data assimilation is the optimization of emission by introducing emission variables in the control vector.*

=> Disagree. And this question is overlapped with #13 above (in yellow). This paper is not meant to examine the uncertainties of emissions, nor the impact of assimilation depending on the lifetime of each aerosol species. Introducing emission variables in the control vector is a whole different topic, warranting another paper. No changes.

*16. When referring to model level in the text or Figure caption it would be helpful to indicate the corresponding elevation (e.g this is missing in line 310 and in the caption of Fig 9)*

=> As stated in the first paragraph of Section 2.1, the elevation of the lowest model level is 173 meters in domain 2. Here, surface PM concentrations were assimilated by computing the model concentrations at the lowest model level, **regardless of the model elevation**. Therefore, it is not necessary to mention the model elevation in Figure 9. No changes.

*Figure and Table specific comments*

*- More details and references could be added in Table1*

=> Added.

*- Table 2 provides very technical information, I wonder if it is relevant for the paper.*

=> Disagree. These are the **new parameters** introduced by the new development. It is important to list the new parameters added into the new system. No changes.

*- Figure 3: the colour legend is missing*

=> Figure 3 is now updated with contours filled in color. And the caption is also modified as "~contouring from 0.1 to 0.9 every 0.1 in different colors" by adding "in different colors" at the end.

*- Figure 4: y axis label, do you mean vertical level ?*

=> Figure 4 is now modified to have "Model level" as a new y-axis label.

*- Figure 5: Does the shaded area in light blue indicate one or two standard deviation ?*

=> When we say standard deviation along with mean, it means one standard deviation.

*- Figure 6 and 7: in the legend caption I suggest to replace omb, oma by o-b, o-a. The representation of the total number of stations is not very informative, I suggest to remove it for clarity. I would suggest keeping only Figure 6.*

=> The caption is now changed to have observation-minus-background(omb; dotted gray line) and observation-minus-analysis (oma; solid black line)". Also, it is good to track down the number of observations assimilated through cycling, which can support the reliability of the analysis system. No changes in the figure.

- *Figure 12: Can you comment on the peaks of the bias shown by both DA and noDA experiments at 12 and 36h forecast time for domain 1 ?*

=> The systematic error peaks might be associated with the forcing parameters such as emissions or boundary forcings, but it is uncertain and not directly related to the implementation of the new system presented in this work. No changes.